# Effect of Topical Programmed Death-Ligand1 on Corneal Epithelium in Dry Eye Mouse

**DOI:** 10.3390/biom14010068

**Published:** 2024-01-04

**Authors:** Ko Eun Lee, Seheon Oh, Basanta Bhujel, Chang Min Kim, Hun Lee, Jin Hyoung Park, Jae Yong Kim

**Affiliations:** 1Moon’s Eye Clinic, Suwon 16200, Republic of Korea; skymeun@naver.com; 2Department of Ophthalmology, University of Ulsan College of Medicine, Asan Medical Center, Seoul 05505, Republic of Korea; basantabhujel86@gmail.com (B.B.); yhun777@hanmail.net (H.L.); 3Research Institute for Biomacromolecules, University of Ulsan College of Medicine, Asan Medical Center, Seoul 05505, Republic of Korea; 72004218osh@gmail.com (S.O.); kcm8821@naver.com (C.M.K.); 4Department of Medical Science, University of Ulsan Graduate School, Seoul 05505, Republic of Korea; 5MS Eye Clinic, Seongnam 13640, Republic of Korea

**Keywords:** programmed death-ligand 1, dry eye disease, desiccation stress, corneal epithelial

## Abstract

Dry eye disease (DED) is a growing health concern that impacts millions of individuals every year, and is associated with corneal injury, excessive oxidative stress and inflammation. Current therapeutic strategies, including artificial tears and anti-inflammatory agents, are unable to achieve a permanent clinical cure due to their temporary nature or adverse side effects. Therefore, here, we investigated the effectiveness of the topical administration of programmed death-ligand 1 (PD-L1) in the mouse model of DED. The model was generated in C57BL/6 mice by excising the extra orbital lacrimal gland and causing desiccation stress with scopolamine injections. Subsequently, either phosphate-buffered saline (3 µL/eye) or PD-L1 (0.5 µg/mL) was topically administered for 10 days. Tear volume was evaluated with phenol red thread, and corneal fluorescein staining was observed to quantify the corneal epithelial defect. Corneas were collected for histological analysis, and the expression levels of inflammatory signaling proteins such as CD4, CD3e, IL-17, IL-1β, pIkB-α, pNF-kB and pERK1/2 were assessed through immunofluorescence and Western blot techniques. Our results demonstrate that desiccating stress-induced corneal epithelial defect and tear secretion were significantly improved by topical PD-L1 and could reduce corneal CD4+ T cell infiltration, inflammation and apoptosis in a DED mouse model by downregulating IL-17 production and ERK1/2-NFkB pathways.

## 1. Introduction

Dry eye disease (DED) is defined in the 2017 Dry Eye Work Shop II (DEWS II) as a multifactorial disease of the ocular surface regarded as a causative mechanism of tear film instability, tear osmotic pressure, inflammation, damage to the eye surface (cornea and conjunctiva) and sensory nerve abnormalities on the eye surface [1]. When tear osmolarity increases owing to any desiccation stress, inflammation is caused by stress signals, and CD4+ T cells are activated [2]. This causes abnormalities in the barrier function of the tear film, decreases secretion and promotes apoptosis, causing instability of the tear film and ultimately forming a vicious cycle [3].

When desiccation stress occurs in the cornea, the tear film barrier collapses, interleukin-17 (IL-17) and interferon gamma (IFN-γ) increase and various inflammatory cytokines are secreted. In addition, goblet cells are decreased in the conjunctiva, CD4+ T cells infiltrated to the cornea [3]. The number of CD4+ T cells was found to increase in the conjunctival epithelium when desiccating stress was applied [2]. The cornea, conjunctiva and tears showed increased levels of IL-17 post-desiccating stress [4]. IL-17 neutralization decreased the levels of matrix metalloproteinases-3 (MMP-3) and matrix metalloproteinases-9 (MMP-9) and associated gelatinolytic activity in the corneal epithelium, and the dysfunction observed in the corneal epithelium was reduced [4]. Furthermore, Chauhan et al. reported that corneal fluorescein staining scores were lower in mice treated with anti-IL-17 antibodies [5]. IL-17 activates mitogen-activated protein kinases (MAPKs) and the P38/extracellular signal-regulated kinase (ERK)/MAPK signaling pathway and is involved in the development of inflammatory skin diseases, including psoriasis [6,7,8].

Programmed cell death protein-1 (PD-1) and its ligand, PD-L1, are negative co-stimulatory molecules that are essential for regulating both physiological and pathological immune responses. PD-1/PD-L1-mediated signaling pathways have been extensively studied in cancer research and are becoming crucial for biopharmaceuticals and immunotherapy [9,10]. The T-cell apoptosis mediated by PD-L1 has only been observed in immune-privileged tissues or sites such as tumors, liver and corneas [11,12]. Various immune checkpoint molecules are constitutively expressed in the cornea, and they regulate immune responses to prevent inflammation-mediated corneal tissue damage [13]. PD-L1 is constitutively expressed on endothelial cells of the cornea, some stromal cells, iris–ciliary body and the neural retina [14,15,16]. The rejection reaction after corneal transplantation is enhanced by the blockade of PD-L1 or PD-1 with antibodies. The T-cell apoptosis mediated by PD-L1 has only been observed in immune-privileged tissues or sites such as tumors, liver and corneas [17].

Although PD-1/PD-L1-mediated signaling pathways have recently been shown to have a close relationship with retinal diseases, there are limited studies on ocular surface disease. In vivo studies, conducted with mice lacking PD-L1 and treated with a PD-L1 blocking antibody, have shown proof supporting the inhibitory roles of PD-L1 in both autoimmune and alloimmune responses [18]. Similarly, a study on murine corneas showed the expression of PD-L1 lowered the risk of corneal allograft rejection by inhibiting autoreactive and alloreactive T cells [19]. Likewise, the expression of PD-L1 on normal and inflamed human ocular cells showed suppression of immune-mediated inflammation by modulating proinflammatory cytokines and Th2 cytokine production by activated T cells [20]. Further, the expression of PD-L1 on murine corneas was also found to play a role in downregulating local immune response and protecting the corneal endothelial cells from killing by T cells [21]. Recent reports indicated that in vitro, retinal pigment epithelial (RPE) cells exposed to IFN-γ express PD-L1 led to the suppression of IFN-γ production by T cells activated with anti-CD3 antibodies [22]. One study demonstrated that various cytokines, like IFN-γ and IL-17, elevated the PD-L1 expression on RPE cells, augmenting their inhibitory impact on pathogenic T cells through PD-L1/PD-1 interaction. This led to a decrease in inflammation mediated by T cells [23].

In the realm of dry eye therapeutics, our investigation focuses on the promising checkpoint inhibitor PD-L1, while drawing insights from established treatments like cyclosporine A, a well-known calcineurin inhibitor which modulates the immune response by inhibiting T cell activation and cytokine release. It forms a complex in T cells, blocking the transcription of cytokine genes, including IL-2 and IL-4, inhibits p38 and JNK activation and ultimately suppressing IL-2 production [24,25,26,27].

Emerging formulations of cyclosporine A, like aqueous nanomicellar formulations, tackle topical administration challenges. These innovations may reduce symptom relief time, enhance tolerability and improve patient adherence. The future of dry eye treatment is anticipated to shift towards noninvasive sustained-release cyclosporine A formulations, ensuring controlled, long-acting therapy for patients [28]. When considering similar drugs, PD-L1, a checkpoint inhibitor, modulates T cell activity by interacting with PD-1, thereby regulating immune responses. While differing in their modes of action, both cyclosporine A and PD-L1 share the overarching goal of addressing immune dysregulation underlying DED.

Thus, this study aimed to investigate the effects of topical PD-L1 on DS-induced corneal epithelial apoptosis and inflammation associated with CD4+ T cells and IL-17 in in vivo mouse models of DED.

## 2. Methods

### 2.1. Animal Model

A total of 36, 8 to 12-weeks-old female C57BL/6 mice were used in this study and were acclimated for 1 week. All animal experiments were performed according to the ARVO Announcement on Animal Use in Ophthalmology and Visual Studies and were reviewed and authorized by the Animal Care and Use Committee of the Life Sciences Institute of Asan Medical Center. The committee complies with the Laboratory Animal Resources Guidelines.

### 2.2. Lacrimal Gland Excision

Unilateral lacrimal gland excision was performed under 1.5% to 2% isoflurane anesthesia. The extraorbital lacrimal gland was accessed through a 3-mm incision made anterior and ventral to the ear, taking care to avoid surrounding blood vessels and nerves [29]. For sham surgeries, an incision was made and the extraorbital lacrimal gland was partially exposed and sutured.

### 2.3. DED Mouse Model

To establish a DED mouse model, we administered subcutaneous injections of scopolamine (0.5 mg/0.2 mL) four times a day for 10 days to mice undergoing lacrimal gland excision. The mice were divided into two groups: the desiccation stress group (*n* = 12) in which phosphate-buffered saline (PBS) was administered; the PD-L1 group (*n* = 12) in which PD-L1, recombinant mouse B7-H1 (PD-L1, CD274)-Fc Chimera (carrier-free; catalog No. 758206, Biolegend, Inc., San Diego, CA, USA) was topically administered. Finally, the control group in which dry eye was not induced (*n* = 12). Topical PD-L1 at a concentration of 0.5 mg/mL or PBS was administered 4 times a day for 10 days.

### 2.4. Tear Volume Measurement

Tears were measured by inserting cotton phenol red threads (Zone-Quick; FCI Ophthalmic, Pembroke, MA, USA) into the lateral canthus of the eye for 15 s in unanesthetized animals. The length of thread presenting with a change in color was measured by caliper (monos) [30].

### 2.5. Corneal Epithelial Fluorescein Staining

Corneal fluorescein staining was performed prior to the induction of desiccation stress, at 5 days and again at 10 days. A 1% fluorescein solution (10 μL; Sigma-Aldrich Corp., St. Louis, MO, USA) was applied to the corneas in isoflurane-anesthetized mice. After 2 min, the eye was rinsed with artificial tears to remove excess fluorescein and examined using cobalt blue light with a handheld ophthalmic slit-lamp (Hai Laboratories, Inc., Lexington, MA, USA). Punctate corneal staining was quantified using ImageJ software (version 1.62f; available by ftp at http://rsb.info.nih.gov/nih-image; accessed on 10 September 2023; developed by Wayne Rasband, National Institutes of Health, Bethesda, MD, USA). The amount of epithelial deficiency was compared between the groups using the ratio of the defect area to the whole corneal area.

### 2.6. Immunofluorescence Staining

Mice were sacrificed after anesthesia with a mixture of tiletamine and zolazepam (Zoletil^®^ 50; Virbac, Inc., Carros, France), and their eyeballs were harvested and fixed in Davidson’s fixative for 8 h followed by 4% paraformaldehyde overnight. The whole corneal tissue was paraffin-blocked. The tissue was then cut to hematoxylin and eosin (H&E) or used for immunostaining. After dewaxing, rehydration and antigen retrieval, the sections were blocked with 0.1% BSA block solution containing normal goat serum for 30 min at room temperature and incubated at 4 °C overnight with antibodies against CD4 (1:200, Catalog No. NBP1-19371; Novus Biologicals, LLC, Centennial, CO, USA), IL-17 (1:100, Catalog No. ab79056; Abcam, Inc., Cambridge, MA, USA) and pERK (1:200, Catalog No. 4370; Cell Signaling Technology, Inc., Danvers, MA, USA). The sections were washed with PBS and incubated with Alexa Fluor 488 (1:1000, Catalog No. A11008; Invitrogen, Inc., Carlsbad, CA, USA) or Alexa Fluor 555 (1:1000, Catalog No. A21424; Invitrogen, Inc.) for 1 h in the dark at room temperature. In addition, the sections were counterstained with DAPI for 5 min and subjected to immunohistochemical staining for PD-L1 (1:50, Catalog No. ab233482; Abcam, Inc.) and CD3e (1:50, Catalog No. MA5-14524; Invitrogen, Inc.). After immunostaining of the tissue slides, 6 samples were randomly selected from each group, and 5 fields of view were randomly selected from each sample. Finally, images were taken under a confocal microscope (Carl Zeiss, Inc., Jena, Germany) or an optical microscope (Olympus, Inc., Tokyo, Japan). The intensity of the images was analyzed using ImageJ software (version 1.62f; available by ftp at http://rsb.info.nih.gov/nih-image; accessed on 10 September 2023; developed by Wayne Rasband, National Institutes of Health, Bethesda, MD, USA).

### 2.7. TUNEL Assay

The cut epithelial tissue sections were baked in an oven at 60 °C for 30 min, followed by dewaxing (5 min, 3 times) with xylene and dehydration with 100% ethanol, 95% ethanol and 70% ethanol (3 times). Terminal deoxynucleotidyl transferase (TdT) deoxyuridine triphosphate (dUTP) nick-end labeling (TUNEL) (684817910; Roche, Inc., Mannheim, Germany). After the nucleus was stained with DAPI, images were taken under a confocal microscope (Carl Zeiss, Inc.) and the number of cells was counted.

### 2.8. Western Blotting

Each corneal sample was washed with cold PBS and incubated on ice for 5 min in 15 of lysis buffer (Cell Signaling Technology, Inc.) containing phosphatase and protease inhibitors and PMSF. Sonication of the tissue lysates was briefly performed with ultrasonic splitting (Sonics Vibra-Cell; Artisan Technology Group, Champaign, IL, USA), and the samples were homogenized on ice 5 times (10 s each with intervals of 5 s). The homogenates were centrifuged at 14,000× *g* for 10 min at 4 °C. The protein concentration of each supernatant was determined using BCA Protein Assay Kit (Thermo Fisher Scientific, Inc., Waltham, MA, USA). These samples were subjected to electrophoresis on 10% or 12% SDS gels, and the proteins were transferred to PVDF membranes (MilliporeSigma, Inc., Burlington, MA, USA) by electroblotting. The membranes were blocked by incubation with 5% BSA in Trisbuffered saline containing Tween 20 TBST for 1 h at room temperature and incubated overnight at 4 °C with specific rabbit polyclonal antibodies against CD4 (1:1000, Catalog No. NBP1-19371; Novus Biologicals, Littleton, CO, USA), IL-17 (1:1000, Catalog No. ab79056; Abcam, Inc.), phosphorylated NF-kB (pNF-kB; 1:1000, Catalog No. 3033S; Cell Signaling Technology, Inc.), NF-kB (1:1000, Catalog No. 51-0500; Invitrogen, Inc.), phosphorylated IkB-α (pIkB-a 1:1000, Catalog No. 9246S; Cell Signaling Technology, Inc.), IkB-α (1:1000, Catalog No. PA5-17888; Invitrogen, Inc.), Bax protein (1:1000, Catalog No. sc-20067; Santa Cruz Biotechnology, Inc., Dallas, TX, USA) and β-actin (1:10,000, Catalog No. 5125S; Cell Signaling Technology, Inc.). The membranes were washed 3 times and incubated for 1 h at room temperature with an HRP-linked secondary antibody followed by visualization of the bands with a chemiluminescence system (WBKLS0100; MilliporeSigma, Inc.). All images were captured, and densitometry analyses were performed using ImageJ software. The levels of expression of the above proteins were normalized to those of β-actin in the same samples.

### 2.9. Statistical Analysis

For the statistical analysis of the data, GraphPad Prism (version 5.01, GraphPad Software, Boston, MA, USA) was used, and ImageJ software (Version 1.50b, https://imagej.nih.gov/ij/; accessed on 10 September 2023; developed by Wayne Rasband, National Institutes of Health, Bethesda, MD, USA was used for the quantification of data. Data are presented as mean, ±standard deviation (SD) and one-way analysis of variance (ANOVA). The level of significance was set as *p* < 0.05.

## 3. Results

### 3.1. Effects of Topical PD-L1 on Corneal Epithelial Changes in Mice with Desiccation Stress-Induced DED

In vivo, we established a DED mouse model via the excision of the extraorbital lacrimal gland and subcutaneous scopolamine injection in the dry environment. Following the development of the dry eye mouse model, fluorescein staining was used to measure the epithelial defect areas in the control, desiccation+PBS and desiccation+PD-L1 group at days 0, 5 and 10. Previous data revealed a significant increase in the fluorescein staining in the corneas of mice exposed to desiccation stress [31,32]. No significant differences between the three groups were observed at the baseline (day 0). However, on day 10, the corneal defect area in desiccation+PBS group was 39%, which was significantly higher than that of the control group (18%). Notably, in the desiccation+PD-L1 group, the expression was 19%, which was similar compared with that of the control group (18%). It showed significant improvement in the desiccation+PD-L1 compared to the desiccation+PBS group (Figure 1A,B).

In addition, in the measurement of tear volume using phenol red thread, the desiccation+PBS group was 0.4 ± 0.2 mm, which was significantly lower than the control group’s 1.3 ± 0.5 mm at day 10 (Figure 1C). The desiccation+PD-L1 group also showed 8.2 ± 0.5 mm, which was less than the control group but significantly improved compared to the desiccation+PBS group. These findings demonstrated that topical PD-L1 administration has a protective effect on the cornea against desiccation stress.

### 3.2. Effects of Topical PD-L1 on Corneal Histological Changes in Mice with Desiccation Stress-Induced DED

We then assessed H&E staining to demonstrate the corneal morphological alterations. Our results for H&E staining showed that in the control group, the superficial corneal epithelial cells (CECs) in the corneas were organized and tightly bound, resulting in a smooth corneal surface (Figure 2). In contrast, the corneas of the desiccation+PBS group, exhibited disruption in the epithelial layers, with desquamation of epithelial cells. However, they were improved in the desiccation+PD-L1 group. Like our study, numerous other studies have shown the abnormal structure of the corneal epithelium in DED groups [32,33,34]. The observation of marked histomorphological differences between the desiccation+PBS and desiccation+PD-L1 groups suggest the future potential of topical PD-L1 in treating DED conditions.

### 3.3. Expression of Corneal PD-L1 and T-Cell Markers in Mice with Desiccation Stress-Induced DED

Next, we performed immunofluorescence staining to investigate the expression of PD-L1, CD4, CD3e and IL-17 localized in the mice corneas (Figure 3A). In our present study, PD-L1 expression was significantly reduced in the DED group compared to the control group. Interestingly, in the dessication+PD-L1 group, its expression was partially restored (Figure 3A,B) indicating that desiccation stress led to a depletion in PD-L1, but topical PD-L1 administration could attenuate desiccation stress-induced reduction in PD-L1 in the corneas of mice with DED.

Prior investigation indicated that desiccating stress led to an increase in the expression of IL-17 and CD4 in the corneas [4,35]. In our study, we observed a substantial elevation in the localized expression of CD4, IL-17 and CD3e in the corneas of the desiccation+PBS group compared to the control group. Conversely, the desiccation+PD-L1 group exhibited significantly reduced expression of these markers (* *p* < 0.05); (Figure 3A(ii–iv),B,D,E). Furthermore, we conducted Western blot analyses to evaluate the expression levels of CD4 and IL-17, and similar changes were observed (Figure 3F–I).

Hence, these results demonstrated that the topical PD-L1 administration could suppress T-cell infiltration in the corneas of mice subjected to desiccation stress.

### 3.4. Effects of Topical PD-L1 on Corneal Inflammation in Mice with Desiccation Stress-Induced DED

To further investigate the alterations in inflammatory-associated factors following topical administration of PD-L1 in the cornea, we performed immunofluorescence staining for IL-1β and pERK1/2. Several studies demonstrated the expression of IL-1β and pERK1/2 were elevated in the desiccation stress-induced DED [4,31,36,37]. In our study, immunofluorescence staining of the cornea showed that the expression of IL-1β, localized in the cornea, was increased in the desiccation+PBS group compared to the control group; however, their expression was partially restored in the desiccation+PD-L1 group (Figure 4A(i),B).

In addition, our findings indicated that the expression of pERK1/2 was significantly increased in the desiccation+PBS group compared to the control group, but markedly reduced in the desiccation+PD-L1 group; however, among them all, the lowest expression was observed in the control group (Figure 4A(ii),C). Hence, these findings indicated that desiccation stress significantly increased the expression of IL-1β and pERK1/2; however, topical PD-L1 administration significantly decreased the inflammatory responses.

Furthermore, we demonstrated the inflammatory response in mouse corneas through pIkB-α and pNF-kB pathways. Previous studies have indicated that the dysregulation of these pathways was activated in the DED mouse model and induced inflammatory responses [36]. In our present study, Western blotting showed that the expression of pIkB-α and pNF-kB in the corneas was increased in the desiccation+PBS group compared to the control group; however, their expression was reduced in the desiccation+PD-L1 group, (Figure 4D–G); thus, suggesting that desiccation could activate the inflammatory markers pIkB-a and pNF-kB and that topically administered PD-L1 could inhibit inflammation in the corneas of mice under desiccation stress.

### 3.5. Effects of Topical PD-L1 on Corneal Apoptosis in Mice with Desiccation Stress-Induced DED

One of the major consequences of DED is an elevated presence of apoptosis markers within the corneal epithelium, indicating heightened cellular apoptosis in the cornea [38,39]. In our study, few TUNEL-positive CECs were detected in the control group; however, many TUNEL-positive CECs were found in the desiccation+PBS group. But, interestingly, only a few TUNEL-positive CECs were found in the desiccation+PD-L1 group (Figure 5A,B).

Further, we performed Western blot analysis for Bax to demonstrate the involvement of the intrinsic apoptotic pathway in DED. Our findings for Western blotting revealed that the expression of Bax was significantly increased in the desiccation+PBS group compared to the control group; however, it was markedly suppressed in the desiccation+PD-L1 group (Figure 5C,D). These findings suggest that topical PD-L1 administration could be a suitable candidate in treating DED and markedly reverse apoptosis in the desiccated CECs.

## 4. Discussion

In the present study, as PD-L1 was reduced in the corneas of mice with desiccation stress-induced DED, we hypothesized that topically administered PD-L1 may prevent T cells from attacking the corneal epithelium and that apoptosis and inflammation may be reduced when topical PD-L1 binds to PD-1. The change in PD-L1 expression was investigated in the corneal tissues of a DED model, in which inflammation was activated by CD4+ T cells. These cells have been found to play a key role in allergy and autoimmunity. The activation of IL-1β, pERK1/2, pNF-ĸB and pIĸB-α was observed in the DED mouse model, with mechanisms similar to those in psoriasis. Moreover, changes in the levels of secreted IL-17 were observed in PD-L1-treated mice with DED.

In this study, the relationship between desiccation stress and T-cell infiltration was determined, and the effect of PD-L1 on T-cell infiltration was confirmed in the desiccation stress-induced DED mouse model. The expression of PD-L1 was greatly decreased in the corneas of mice with DED, which was consistent with the findings of a previous study [18]. Notably, we showed that topically administered PD-L1 prevented T-cell infiltration via the suppression of IL-17, led to the reversal of corneal desiccation stress-induced alterations, and reduced the expression of pro-inflammatory and pro-apoptotic proteins. Tissue-specific PD-L1 expression has been reported to protect against autoimmune diabetes, ocular inflammation and corneal allograft rejection by inhibiting autoreactive and alloreactive T cells [19,20,40,41]. El Annan et al. demonstrated that decreased PD-L1 expression was associated with increased T-cell (both Th1 and Th2) chemokine expression, increased T-cell infiltration and increased corneal fluorescein staining in PD-L1^−/−^ mice and mice treated with antiPD-L1 blocking antibodies [41]. However, thus far, it has remained unclear whether topically administered PD-L1 can modulate T-cell infiltration in DED and regulate inflammation and apoptosis caused by T-cell infiltration. In the present study, we analyzed CD3e and CD4 expression, demonstrating that CD3e and CD4 were markedly increased in DED; however, they were significantly suppressed by topical PD-L1. The findings indicated that topically administered PD-L1 may have immunomodulatory effects on the infiltration of CD4+ T cells.

There have been several previous studies of the immunopathogenesis of DED with a focus on the inflammatory milieu of the tear film or conjunctiva. Chen et al. measured the mRNA levels of IFN-γ and IL-17 in the conjunctiva of C57BL/6 mice with DED induced by scopolamine in a controlled environmental chamber; their levels were elevated in acute DED (day 14), and IFN-γ was normalized in chronic DED (day 126); however, IL-17 levels remained elevated [42]. Nevertheless, corneal inflammation is the most clinically recognizable and important ocular manifestation of DED [43,44,45,46,47,48,49].

The excessive activation of the MAPK pathway is a common cause of many diseases including autoimmune diseases, and sustained ERK phosphorylation activates T helper 17 (Th17) cells [50,51]. Therefore, we examined the effects of topically administered PD-L1 on the expression of pERK1/2 and IL-17 production by Th17 cells in the desiccation stress-induced DED mouse model, which revealed that IL-17 expression by CD4+ T cells was markedly increased in DED and that ERK1/2 was activated; however, they were significantly suppressed by topical PD-L1. Therefore, the ERK signaling pathway may be associated with DED.

CD4+ T-cell-mediated inflammation is known to play a critical role in the pathogenesis of DED [2,5,52,53,54,55]. The production of pro-inflammatory cytokines is closely regulated by the activation of several signaling pathways, including NF-kB and MAPKs (ERK1/2, JNK and p38). To further clarify the underlying mechanisms involving these signaling proteins, we assessed the effects of topical PD-L1 on MAPK and NF-kB activation. We found that ERK1/2 and pNF-kB were activated in mouse corneas under DS and that ERK1/2 and pNF-kB activation could be weakened in mice treated with topical PD-L1. These findings indicated that the anti-inflammatory effects of topical PD-L1 may be attributed to the inhibition of the activation of the MAPK and NF-kB signaling pathways. Furthermore, PD-L1 suppressed the activation of the upstream molecule IkB-α.

Patients with DED are often presented with a certain degree of CEC apoptosis, inflammatory cell infiltration, and even the tendency of CECs to transform into small cells [56,57]. These pathological processes will ultimately lead to the apoptosis of CECs [58]. Therefore, the inhibition of CEC apoptosis may be regarded as a treatment target for DED. Zhang et al. observed that DS significantly increased IFN-γ and IL-17 production by CD4+ T cells in a murine model and found that in vivo neutralization of INF-γ alleviated CEC apoptosis induced by adoptively transferred CD4+ T cells [58]. However, the precise role of IL-17 in inducing CEC apoptosis under desiccation stress has not been established. In the present study, the role of IL-17-producing CD4+ T cells in CEC apoptosis was investigated. Although the effect of PD-L1 on DED remains unclear, we hypothesized that topical PD-L1 may have a beneficial effect on CEC apoptosis by suppressing IL-17 production in DED. Furthermore, the effect of PD-L1 on the expression of the pro-apoptotic protein Bax in the corneas of mice with DED was assessed by Western blotting. Apoptotic CECs were quantified by TUNEL staining. It was found that PD-L1 suppressed Bax upregulation at the protein level. TUNEL staining also revealed that the number of apoptotic CECs in the DED group was higher than that in the blank control group. After topical administration of PD-L1, the number of apoptotic CECs was markedly reduced. Therefore, desiccation stress could induce ERK1/2 and IL-17 production by CD4+ T cells and promote inflammation and CEC apoptosis, and topical PD-L1 could prevent desiccation stress-induced ERK1/2 activation and IL-17 production and alleviate CEC apoptosis and inflammation, which occur in DED.

## 5. Conclusions

In conclusion, topical PD-L1 could reduce corneal CD4+ T cell infiltration, inflammation and apoptosis in a DED mouse model. The protective mechanisms might be related to the PD-1/PD-L1 axis with the downregulation of IL-17 produced by CD4+ T cells and inhibition of the MAPK and NF-kB signaling pathways. Therefore, topical PD-L1 could be a possible candidate for the treatment of DED. Moreover, the multifaceted advantages of PD-L1, including targeted immunomodulation, efficacy in addressing inflammation, stability, cost-effective and its favorable formulation characteristics, position it as a promising and practical therapeutic option for DED. Thus, in the future, utilizing simple topical applications of PD-L1 might be a preferable option to non-biological medications and alternative approaches for treating DED.

## Figures and Tables

**Figure 1 biomolecules-14-00068-f001:**
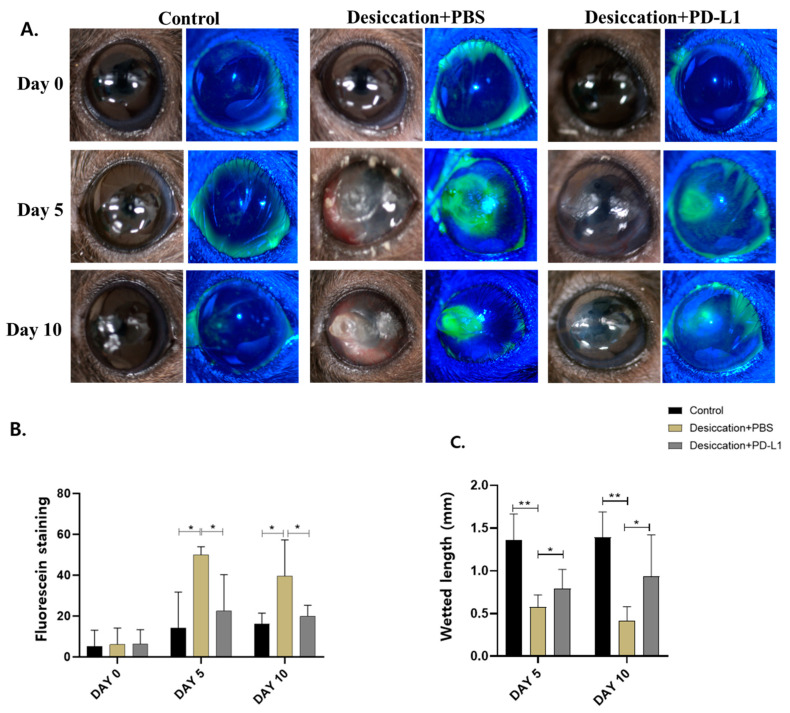
Comparisons of the PD-L1 for DED mouse corneal epithelial recovery. (**A**) Fluorescein-stained mice corneas in control, desiccation+PBS and dessication+PD-L1 group were assessed by slit-lamp microscopy on days 0, 5 and 10 (**B**) Corneal fluorescein staining in control, desiccation+PBS and desiccation+PD-L1 group was presented with a bar graph. (**C**) Tear volume rate (mm wetted in 15 s) in a DED mouse model over 10 days. (* *p* < 0.05; ** *p* < 0.01; significant difference by one-way ANOVA).

**Figure 2 biomolecules-14-00068-f002:**
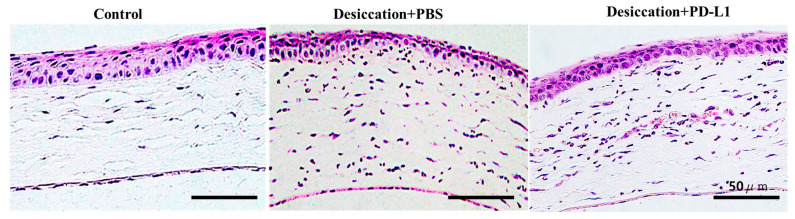
Changes in the corneal morphological alterations in control, desiccation+PBS and desiccation+PD-L1 group. H&E staining showed that superficial CECs in the control group were well arranged and tightly attached and the corneal surface was smooth. The corneal epithelial layers were disrupted in the desiccation+PBS, and were improved in the desiccation+PD-L1 group (*n* = 6, day 10).

**Figure 3 biomolecules-14-00068-f003:**
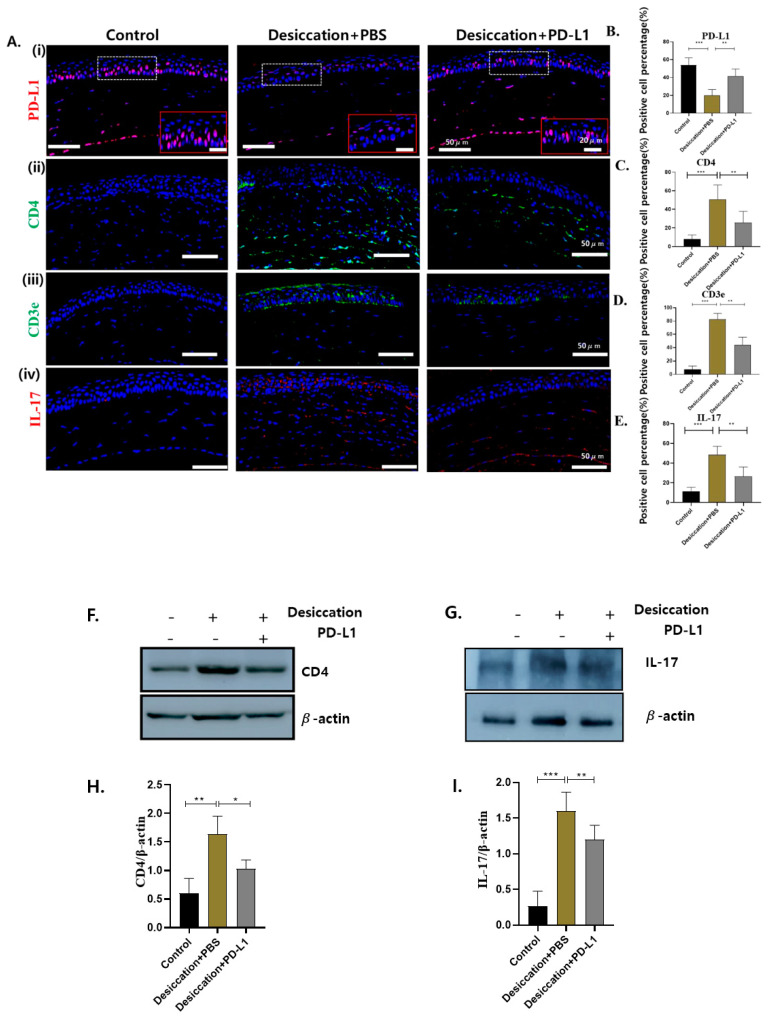
(**A**) Immunofluorescence staining showing the expression of (i) PD-L1, (ii) CD4, (iii) CD3e and (iv) IL-17 localized in the corneas in the control, desiccation+PBS and desiccation+PD-L1 group. (**B**–**E**) Changes in the percentage of PD-L1, CD4, CD3e and IL-17 positive cells in the cornea. Immunopositivity for PD-L1 is shown in higher-power fields. Immunopositivity was counted in low-power fields and calculated as relative to the total number of DAPI-positive cells. (**F**,**G**) Western blot showing the expression of CD4 and IL-17 in the cornea. (**H**,**I**) Relative expression ratio of CD4 and IL-17 to β-actin. In (**B**–**E**,**H**,**I**), data are presented with bar graph (*n* = 6, day 10, * *p* < 0.05; ** *p* < 0.01; *** *p* < 0.001 significant difference by one-way ANOVA). Original images of (**F**,**G**) can be found in Appendix A.

**Figure 4 biomolecules-14-00068-f004:**
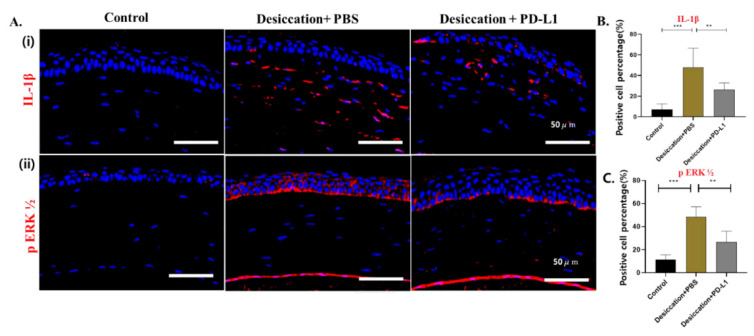
(**A**) Immunofluorescence staining showing the expression of IL-1β (i) and pERK1/2 (ii) localized in the corneas in the control, desiccation+PBS and desiccation+PD-L1 group. (**B**,**C**) Changes in the percentage of positive cells of IL-1β and pERK1/2 in the cornea. Immunopositivity was counted in low-power fields and calculated as relative to the total number of DAPI-positive cells. (**D**,**E**) Western blot showing the expression of pNF-Kb and pIkB-alpha in the cornea. (**F**,**G**) Relative expression ratio of pNF-kB and pIkB-αto β-actin. In (**B**,**C**,**F**,**G**), data are presented with bar graph (*n* = 6, day 10, * *p* < 0.05; ** *p* < 0.01; *** *p* < 0.001 significant difference by one-way ANOVA). Original images of (**D**,**E**) can be found in Appendix A.

**Figure 5 biomolecules-14-00068-f005:**
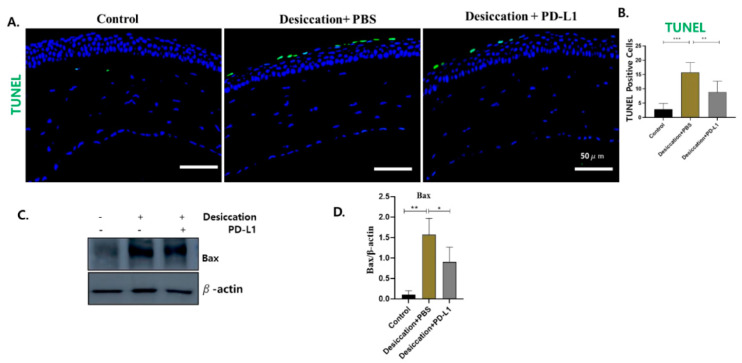
(**A**) TUNEL staining in control, desiccation+PBS and desiccation+PD-L1 group in the cornea. (**B**). Changes in the number of TUNEL positive cells in the cornea. (**C**) Western blot showing the expression of Bax in the cornea. (**D**) Relative expression ratio of Bax to β-actin. In (**B**,**D**), data are presented with bar graph (*n* = 6, day 10, * *p* < 0.05; ** *p* < 0.01; *** *p* < 0.001 significant difference by one-way ANOVA). Original images of (**C**) can be found in Appendix A.

## Data Availability

All datasets used and/or analyzed during the current study are available from the corresponding author on reasonable request.

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
