# Peer review of "Effect of Topical Programmed Death-Ligand1 on Corneal Epithelium in Dry Eye Mouse"

_biomolecules, 2024, doi:10.3390/biom14010068_

Round 1
Reviewer 1 Report
Comments and Suggestions for Authors
Introduction needs to present more studies in details about PDL-1 in other ocular conditions and similar drug target the same mechanism or cause of dry eye like sandimmune cyclosporine eye drops.
‘’ 0.5 g/mL or PBS was administered 4 times a day for 10 days’’ please check the units used after concentrations. It is no way to be that massive g/ml for a biological compound.
The dry eye model was highly invasive by removing lacrimal glands and injecting a scopolamine solution. Why you did not use either one of these two methods that would be enough to induce dry eye.
The conclusion should outline advantages of using PDL-1 over other non-biological drugs such as cyclosporine and other small drug molecules which are easy to formulate in ter ms of stability and cost compared to using biological drug.
Comments on the Quality of English LanguageNone
Author Response
Dear Editor-in-chief,
We really appreciate sending the sincere feedback quickly and the comments that reviewers found interesting to our manuscript “Effect of Topical Programmed Death-Ligand1 on Corneal Epithelium in Dry Eye Mouse” (biomolecules-2740661). We have made corrections focusing on the comments from reviewers. In addition to this, we have checked the entire contents of the manuscript and corrected the sentences. Many edits have been made to reduce the ambiguity of the overall content flow and make it easier for readers to understand. A list of changes is highlighted with red in the revised manuscript
Response to Reviewer:
Reviewer 1
Reviewer 1 comments
#1. Introduction needs to present more studies in details about PDL-1 in other ocular conditions and similar drug target the same mechanism or cause of dry eye like sandimmune cyclosporine eye drops.
We are grateful for your insightful comments and suggestions. We appreciate your thorough review of our manuscript. In response to your query regarding additional studies on PDL-1 in other ocular conditions and similar drug targets for dry eye, we have conducted a more in-depth analysis to enrich the context of our research.
We have pointed several pivotal research studies that explore the involvement of PDL-1 in diverse ocular conditions. (Edit line: 68-84, References- 18,19,20,21,22, and 23). Similarly, we have explained and compared the cyclosporine A and their mechanism in treating DED (Edit line: 85-99, References 24,25,26,27, and 28)
#2.‘’ 0.5 g/mL or PBS was administered 4 times a day for 10 days’’ please check the units used after concentrations. It is no way to be that massive g/ml for a biological compound.
We are thankful for bringing up the concern regarding the units used for the concentrations in our manuscript. We appreciate your careful review, and we acknowledge the oversight in reporting the concentration.
Upon reevaluation, it appears there was an error in the units provided. The correct unit for concentrations of biological compounds should indeed be expressed in milligrams per milliliter (mg/mL) rather than grams per milliliter (g/mL). We apologize for any confusion this may have caused. We have made correction in the unit. (Edit line:125, section 2.3)
#3. The dry eye model was highly invasive by removing lacrimal glands and injecting a scopolamine solution. Why you did not use either one of these two methods that would be enough to induce dry eye.
Thank you so much for your thoughtful review and insightful comments regarding our dry eye model. We appreciate your concern about the invasiveness of our chosen methods, specifically the removal of lacrimal glands and the injection of a scopolamine solution.
The removal of lacrimal glands addresses the deficiency in aqueous tear production, while scopolamine induces decreased tear secretion by affecting the muscarinic receptors, contributing to an overall more comprehensive dry eye. The decision to utilize combination method, both the removal of lacrimal glands and the injection of a scopolamine solution was based on existing literature and well-established precedents. Following studies have employed a combination of these methods to induce a more robust and reliable dry eye model. The rationale behind this approach is to mimic multifactorial aspects of dry eye syndrome, considering both aqueous and evaporative components.
References
-Vittitow, J. L., et al. "An Experimental Model of Chronic Dry Eye Combining Extraorbital Lacrimal Gland Excision and Subcutaneous Scopolamine in the Rat." Investigative Ophthalmology & Visual Science 47.13 (2006): 5598-5598.
-Kim, Minha, et al. "Impact of Lacrimal Gland Extraction on the Contralateral Eye in an Animal Model for Dry Eye Disease." Korean Journal of Ophthalmology: KJO 36.4 (2022): 318.
#4. The conclusion should outline advantages of using PDL-1 over other non-biological drugs such as cyclosporine and other small drug molecules which are easy to formulate in terms of stability and cost compared to using biological drug.
Thank you for your constructive feedback on the conclusion of our manuscript. We appreciate your suggestion to highlight the advantages of using PDL-1 over non-biological drugs, particularly in comparison to cyclosporine and other small drug molecules. We understand the importance of addressing factors such as stability and cost in the formulation of therapeutic agents. By addressing the reviewer’s suggestion, we have added more information on the conclusion part. (Edit line: 405-410)
I gratitude for the reviewer’s advice.
With best regards,
Jae Yong Kim, MD, PhD
Professor
Department of Ophthalmology, Asan Medical Center,
University of Ulsan College of Medicine, Seoul 05505, Republic of Korea
Email: jykim2311@amc.seoul.kr
Reviewer 2 Report
Comments and Suggestions for Authors
This paper evaluated the potential therapeutic effects of PD-L in a dry eye disease mouse model in terms of corneal epithelial barrier and tear retention. Modification to the descriptions of the results should be made for clarity.
-Modify the abstract to specify the hypothesis and what was tested. Re-phrase the sentence 'Thus, to overcome this issue we utilize...".
-Fig. 1B - This panel is described in the text as a measure of fluorescein stain. Please modify the title of the y-axis in the graph appropriately given that corneal epithelial cell number were not measured.
-Fig. 2 - Please specify if the images are representative and the total n used in the H&E analysis. Inclusion of the timepoint in the figure legend should be added. The desicaation+PD-L1 image looks like an uninjured control. One would expect some indication of desiccation with this group, especially given the TUNEL data shown in a latter figure.
Comments on the Quality of English LanguageMinor edits and reformatting are needed throughout.
-Please correct typos and grammar issues throughout (ex: Abstract, line 21- fragment sentence).
-Bold lettering in certain places of the text (pg. 9).
Author Response
Dear Editor-in-chief,
We really appreciate sending the sincere feedback quickly and the comments that reviewers found interesting to our manuscript “Effect of Topical Programmed Death-Ligand1 on Corneal Epithelium in Dry Eye Mouse” (biomolecules-2740661). We have made corrections focusing on the comments from reviewers. In addition to this, we have checked the entire contents of the manuscript and corrected the sentences. Many edits have been made to reduce the ambiguity of the overall content flow and make it easier for readers to understand. A list of changes is highlighted with red in the revised manuscript
Response to Reviewer:
Reviewer 2
Comments and Suggestions for Authors
#1. This paper evaluated the potential therapeutic effects of PD-L in a dry eye disease mouse model in terms of corneal epithelial barrier and tear retention. Modification to the descriptions of the results should be made for clarity.
We are grateful for your valuable feedback on our manuscript. We appreciate your suggestion to modify the descriptions of the results to enhance clarity. We have carefully revised the all the results sections with more descriptions.
-Section 3.1 (edit line: 204-212, 222-227)
-Section 3.2 (edit line: 230-239)
-Section 3.3 (edit line: 248-249, 255-263)
-Section3.4 (edit line: 276-292)
-Section 3.5 (edit line: 308-318)
#2. -Modify the abstract to specify the hypothesis and what was tested. Re-phrase the sentence 'Thus, to overcome this issue we utilize...".
Thank you so much for your constructive feedback on our abstract. We appreciate your suggestion to specify the hypothesis and clarify what has been tested. Additionally, we understand the need for rephrasing the sentence related to overcoming the mentioned issue. We have modified the abstract according to your suggestions. (Edit line:15-30)
#3.-Fig. 1B - This panel is described in the text as a measure of fluorescein stain. Please modify the title of the y-axis in the graph appropriately given that corneal epithelial cell number were not measured.
Thank you for your keen observation regarding Fig. 1B, and we appreciate your suggestion to modify the title of the y-axis to accurately reflect the measure of fluorescein stain, considering that corneal epithelial cell number was not measured. We have carefully revised the title for clarity: We have changed the Y-axis into Fluorescein staining.
#4-Fig. 2 - Please specify if the images are representative and the total n used in the H&E analysis. Inclusion of the timepoint in the figure legend should be added. The desicaation+PD-L1 image looks like an uninjured control. One would expect some indication of desiccation with this group, especially given the TUNEL data shown in a latter figure
We are grateful for your valuable feedback on our manuscript. We have added the total (n) in our revised manuscript. (Edit line 244-245). Similarly, we have added timepoint in all our figure legends.
Thank you for your detailed feedback on Figure 2. We appreciate your suggestions to enhance the clarity of the figure. We have changed the figure of Desiccation+PD-L1 group with further more descriptions in the section 3.2.
#5. Comments on the Quality of English Language
Minor edits and reformatting are needed throughout.
-Please correct typos and grammar issues throughout (ex: Abstract, line 21- fragment sentence).
-Bold lettering in certain places of the text (pg. 9).
Thank you so much for your feedback on the quality of the English language in our manuscript. We appreciate your attention to detail. To address this, we have performed a comprehensive review to make minor edits and improvements for clarity and fluency throughout the manuscript.
-We have modified the manuscript.
-We have changed the bold letters and modified the sentence in our revised manuscript. (Edit line: 308-310)
I gratitude for the reviewer’s advice.
With best regards,
Jae Yong Kim, MD, PhD
Professor
Department of Ophthalmology, Asan Medical Center,
University of Ulsan College of Medicine, Seoul 05505, Republic of Korea
Email: jykim2311@amc.seoul.kr
Round 2
Reviewer 1 Report
Comments and Suggestions for Authors
The authors have satisfactorily addressed the comments.
Reviewer 2 Report
Comments and Suggestions for Authors
The revision addressed most of this reviewer's concerns.
Comments on the Quality of English LanguageNone.